# Thin Film Biocomposite Membrane for Forward Osmosis Supported by Eggshell Membrane

**DOI:** 10.3390/membranes12020166

**Published:** 2022-01-30

**Authors:** Teayeop Kim, Sunho Park, Yoonkyung Lee, Jangho Kim, Kyunghoon Kim

**Affiliations:** 1School of Mechanical Engineering, Sungkyunkwan University, Suwon 16419, Korea; skkty@skku.edu (T.K.); yi9257kr@skku.edu (Y.L.); 2Department of Rural and Biosystems Engineering, Chonnam National University, Gwangju 61186, Korea; preference9330@gmail.com; 3BK21 FOUR Center for IT-Bio Convergence System Agriculture, Chonnam National University, Gwangju 61186, Korea

**Keywords:** forward osmosis, biomaterials, desalination, sustainable process

## Abstract

There is a general drive to adopt highly porous and less tortuous supports for forward osmosis (FO) membranes to reduce internal concentration polarization (ICP), which regulates the osmotic water permeation. As an abundant waste material, eggshell membrane (ESM) has a highly porous and fibrous structure that meets the requirements for FO membrane substrates. In this study, a polyamide-based biocomposite FO membrane was fabricated by exploiting ESM as a membrane support. The polyamide layer was deposited by the interfacial polymerization technique and the composite membrane exhibited osmotically driven water flux. Further, biocomposite FO membranes were developed by surface coating with GO for stable formation of the polyamide layer. Finally, the osmotic water flux of the eggshell composite membrane with a low structural parameter (~138 µm) reached 46.19 L m^−2^ h^−1^ in FO mode using 2 M NaCl draw solution.

## 1. Introduction

Forward osmosis (FO), as a novel and emerging separation technique, has received attention for various applications from low-cost water purification [1] and renewable energy production [2] to protein enrichment [3]. Polyamide-based thin film composite (TFC) membranes are the most popularly used membrane type in FO [4]. TFC membranes have two important layers, a thin selective layer that enables osmotic water transport and a porous support layer that confers mechanical stability and enables molecule diffusion. In contrast with pressure driven separation, in the FO process, the transport property of the selective layer may be limited by internal concentration polarization (ICP) due to the support layer.

Higher water transport can be achieved in the FO process by enhancing the water permeability of the selective layer and by structural enhancement of the support layer to reduce the ICP. ICP reduces the effective osmotic pressure applied to the selective layer, as the permeated water dilutes the draw solution (DS) in the support layer.

The effective diffusion length through the support layer pore, which is defined as a structural parameter, affects the degree of ICP [5]. The low structural parameter results weaker ICP and can be achieved by higher porosity, thinner thickness, and lower pore tortuosity. To achieve a low value of the structural parameter corresponding to a lower ICP, a highly porous, thin, and less tortuous structure is needed. In previous studies on minimizing the ICP, structural parameter was reduced by optimizing the material composition [6,7] and incorporating nanoparticles [8,9,10,11] by the phase inversion method, and by employing a novel fabrication strategy using electrospinning [12,13,14]. The most effective results were achieved with a polymeric fibrous support fabricated by electrospinning that decreased the value of the structural parameter and the tortuosity to 66 µm and 1.2, respectively [13].

Despite these developments, contradictory environmental problems remain to be considered. Synthetic polymer materials constituting the FO membrane for water treatment are known to cause water pollution during the synthesis process [15]. Furthermore, additional hazardous organic solvents are used in the manufacturing process for polymeric membranes [16]. Therefore, not only performance of membranes, the environmental issue of material and process should be considered. Compared to these artificially fabricated polymeric supports for FO membranes, the eggshell membrane (ESM) has the required properties. In terms of the structural aspects, the ESM is thin (~100 µm) [17] and comprises an entangled microfibrous structure that is most desirable for reducing the ICP [18]. Furthermore, the unique properties of the ESM, including the flexibility, thermal stability [19], chemical stability to organics [20], and hydrophilic surface [21] enable further functionalization of the ESM layer via thermal and chemical processes for advanced applications. Most importantly, the ESM is derived from affordable waste materials. More than 10^12^ eggs are consumed annually [22], and most of the by-products, including the ESM, are disposed [23]. Therefore, the use of ESM based platforms instead of synthetic polymers leads to the reduction of chemical waste and the development of high-value products by using waste agricultural by-products.

Given the advantages and suitability of the ESM as an FO membrane support, we herein fabricate an ESM-based FO membrane combined with a polyamide selective layer (Figure 1a). Further, an ESM composite with graphene oxide (GO) is developed to overcome the problems encountered in FO membrane fabrication (Figure 1b). The fabricated FO membranes based on ESM and ESM-GO are evaluated in terms of the osmotic water flux and reverse ion flux using NaCl DS of different concentrations.

## 2. Materials and Methods

### 2.1. Materials

Both 1,3,5-Benzenetricarbonyl trichloride (TMC, 98%) and graphite power (~200 mesh, 99.9%) were purchased from Alfa-Aesar. *m*-Phenylenediamine (MPD, 99%), and sodium nitrate (NaNO_3_, 99%), and potassium permanganate (KMnO_4_, 99%) were obtained from Sigma-Aldrich. Sodium chloride (NaCl, 99%), hydrogen peroxide (H_2_O_2_, 30%), sulfuric acid (H_2_SO_4_, 95%), and hydrochloric acid (HCl, 99%) were purchased from Daejung chemicals. Deionized water was produced by EXL-3 water purification system (Vivagen, Seongnam, Korea).

### 2.2. Preparation and Characterization of ESM and ESM-GO Composite

ESM was manually separated from eggshells, and the prepared ESM was washed with DI water several times to remove the yolk and residue (Figure 2a). The separated ESM was cut into 20-mm rounds to control the shape of the sample, and the ESM was placed on a glass substrate. 

To obtain the flat and hydrophilic surface desired for polyamide layer deposition, one side of the prepared ESM was coated with graphene oxide. Graphene oxide was prepared by the modified Hummers method [24]. Briefly, 5 g of graphite and 2.5 g of sodium nitrate were combined with 115 mL of sulfuric acid under constant stirring in an ice bath. After 30 min, 15 g of KMnO_4_ was slowly added to the solution while maintaining the temperature below 20 °C. The mixture was stirred at 30 °C for 30 min and the resulting solution was diluted by adding 230 mL of hot water under vigorous stirring. The solution was further treated with 30% H_2_O_2_ solution (50 mL) and 400 mL of water. The resulting mixture was washed with HCl and H_2_O, respectively. To obtain a clear homogeneous solution, the prepared graphene was dispersed in DI water and sonicated for 2 h before use. 

To obtain the flat and hydrophilic surface desired for polyamide layer deposition, one side of the prepared ESM was coated with graphene oxide. Graphene oxide was prepared by the modified Hummers method [24]. Briefly, 5 g of graphite and 2.5 g of sodium nitrate were combined with 115 mL of sulfuric acid under constant stirring in an ice bath. After 30 min, 15 g of KMnO_4_ was slowly added to the solution while maintaining the temperature below 20 °C. The mixture was stirred at 30 °C for 30 min and the resulting solution was diluted by adding 230 mL of hot water under vigorous stirring. The solution was further treated with 30% H_2_O_2_ solution (50 mL) and 400 mL of water. The resulting mixture was washed with HCl and H_2_O, respectively. To obtain a clear homogeneous solution, the prepared graphene was dispersed in DI water and sonicated for 2 h before use.

As illustrated in Figure 2b, GO solution (0.1, 0.5, and 1%; 140 µL) was uniformly coated on the ESM-glass substrate using a spin-coater (ACE-200) at 7000 rpm for 30 s and dried at room temperature. Finally, fabricated ESM-GO composites were denoted by ESM-GO 0.1, 0.5, 1 following GO contents (1, 0.5, 1%) of solution used in spin coating.

The morphology of the membranes and FO composite membranes was evaluated by field-emission scanning electron microscopy (FESEM, JEOL JSM-7600F). To prepare samples for cross-surface imaging, the membranes were fractured after freezing in liquid nitrogen. The Raman spectra of the membrane surface were acquired using a confocal Raman spectrometer (XperRam Compact, Nanobase, Seoul, Korea). The contact angle (CA) was analyzed with a contact angle analyzer (SDL200TEZD, FEMTOFAB Co., Ltd., Pohang, Korea) by dropping 3 µL of DI water on the sample surface. The pure water permeability was calculated by measuring filtered deionized water through membranes for 10 min, in vacuum filter applied 93 kPa.

### 2.3. Deposition of Polyamide Layer

A polyamide layer was prepared by interfacial polymerization on the membranes as illustrated in Figure 2c. MPD (2 wt%) was dissolved in DI water and 0.1 wt% TMC was dissolved in n-hexane by bath sonication for 1 h. The prepared ESM or GO-ESM composite was immersed in DI water and placed in a desiccator for 1 h to wet the membrane and remove any microbubbles in the membranes. The wet membranes were placed in aqueous MPD solution for 5 min and placed on a flat glass plate. Thereafter, the aqueous MPD solution was removed by blowing with air and TMC-hexane solution was passed over the membrane surface for 90 s. The membranes were instantly washed with pure n-hexane to remove non-reacted TMC and cured in a drying oven at 60 °C for 5 min for stabilization [25] and to enhance interfacial adhesion between polyamide and the substrates. Finally, the fabricated ESM-based FO membranes were placed in DI water in the dark for 24 h to extract MPD from the membranes. The fabricated FO membranes are denoted as eggshell composite (ESC, ESC-GO 0.1 to 1) depending on the use of ESM or the ESM-GO composite membrane (ESM, ESM-GO 0.1 to 1).

### 2.4. Lab-Scale FO Test

The FO performance was evaluated by lab-scale FO tester presented in Figure 3. The FO performance, the prepared FO composite membrane was tightly held between rubbery O-rings with a diameter of 14 mm. The DS was passed over the support layer, and the feed solution (FS) was passed over the polyamide layer with a constant flux of 250 mL min^−1^. DI water was used for FS and NaCl aqueous solution with different concentrations (0.5, 1, 1.5, and 2 M) used for DS. Each FO experiment was conducted for 2 h with each different sample.

The pure water flux (*J_w_*) was characterized based on the volume permeation per unit effective area and the experimental time according to the following equation:$$J_{w}=\frac{\Delta V_{f}}{A_{m}\cdot \Delta t}$$
where, $\Delta V_{f}$ is the volume permeation of FS, *A_m_* is the active surface area of the membrane (1.56 cm^2^), and Δ*t* is the operation time for the experiment.
$$J_{s}=\frac{\Delta \left(c_{f}V_{f}\right)}{A_{m}\cdot \Delta t}$$

Here, *c_f_* and *V_f_* are the concentration and volume of NaCl in the FS, measured before and after the experiment. 

The performance parameter of the membranes, pure water permeability (*A*), ion permeability (*B*), and structural parameter (*S*) and coefficient of determination (*R*^2^) were determined using a analytical method developed by Alberto Tiraferri et al. [26] according to the equation:$$S=\frac{D}{J_{v}}ln\frac{A\cdot \pi_{draw}+B}{A\cdot \pi_{feed}+J_{v}+B}$$
where *π_draw_* and *π_feed_* are the osmotic pressure of the DS and FS calculated using the van’t Hoff equation. *D* is the diffusion coefficient of NaCl in aqueous solution, which was calculated in a previous study [27].

## 3. Results and Discussion

### 3.1. Membrane Characterization

Figure 4 shows the morphology of the cross-surface and top surface of the ESM and ESM-GO membranes imaged by FESEM. Before deposition of GO, the structure of the cross-surface (Figure 4a) and top surface (Figure 4e) was porous, composed of accumulated eggshell fibers randomly oriented. After the GO coating process, the GO flakes gradually cover the ESM surface in proportion to the loading. As GO loading increases, GO flakes partially cover the ESM-GO 0.1, 0.5 (Figure 4f,g) surface and fully cover the ESM-GO 1 (Figure 4d,h). As shown in Figure 4b–d, GO formed a thin layer on the surface of the ESM rather than infiltrating into the eggshell fibers. Compared to the surface of the bare ESM membrane, the surface of the ESM-GO membrane was flattened by GO that accumulated in the aperture between the eggshell fibers.

The pure water permeability was evaluated to investigate the effects of the deposited GO layer on the composite membrane (Figure 5a). The pure water permeability of ESM was as high as 15,400 L m^−2^ h^−1^ bar^−1^, which is considerably higher than that of TFC membrane supports fabricated by phase inversion [8]. This high pure water permeability may be attributed to micro-scale flow paths between the eggshell fibers. After GO deposition, the pure water permeability gradually decreased for ESM-GO 0.1 and 0.5, but decreased dramatically for the ESM-GO 1 membrane. As shown in Figure 4h, the compact coverage of the GO layer over the entire surface of ESM-GO 1 caused a dramatic decline in the pure water permeability. 

The contact angles of the ESM and ESM-GO composites were measured to investigate the hydrophilicity, as shown in Figure 5b. In synthesis of polyamide-based desalination membrane, the hydrophilicity of the supporting layer contributed to enhanced interfacial adhesion with the polyamide layer [28,29,30]. Even before GO coating, the surface of the ESM was hydrophilic, as indicated by the contact angle of 74°, due to the numerous hydroxyl, carboxyl, and amide groups [21]. Nevertheless, the surfaces of the GO-coated ESM composite membranes (ESM-GO 0.1, 0.5, 1) were even more hydrophilic. The contact angle gradually decreased with increasing GO content and finally reached ~39° for ESM-GO 1. These results suggest that more GO coated on the ESM enhanced the hydrophilicity of the ESM-GO composite membrane.

Figure 5c shows the Raman spectra of bare GO and the top surface of the membrane, which were used to investigate the properties of GO and confirm deposition. In the Raman spectra of carbon materials, the G-band represents sp^2^ carbon atoms composing graphene and the D-band (*I_d_*/*I_g_* = 0.86) represents structural disorder, which is inevitable due to oxidation [31]. The Raman spectra (*I_d_*/*I_g_* ratio of 0.86) confirm that GO is composed of oxidized graphene sheets [32]. The Raman spectrum of ESM did not present specific peaks. Therefore, the intensity of the D and G bands of the ESM-GO composite, which is proportional to the GO loading, indicates a deposited GO layer on the membrane surface. Consequently, the Raman spectra of ESM-GO 1 without the broad line induced by ESM indicate a considerably thickened GO layer that compactly covers the membrane surface.

### 3.2. Characterization of FO Membranes

Figure 6 shows the intrinsic morphology of polyamide, described as a “ridge-and-valley” structure [33], which confirms deposition of the polyamide layer on the membrane surface. Compared to the bare ESC membrane (Figure 6a), the ridge-and-valley structure tended to become flatter as the GO content increased (Figure 6b–d). This tendency was observed in the GO/polyamide composite matrix and is attributed to hindrance of MPD diffusion by GO. After interfacial polymerization, the polyamide active layer was located on the eggshell fibers of the ESC membrane or GO layer of the ESC-GO membranes (Appendix A).

As shown in Figure 5, bare ESM has a relatively rough surface composed of protein fibers (~3 µm), compared to the electro-spun fibers (~150 nm) used as the membrane support in previous studies [12,18]. This structural disadvantage led to some obvious defects in the polyamide layer, permitting direct ion transport (Appendix A). However, these hole-shaped defects were not found in ESC-GO 0.1 and 0.5 with the polyamide layer on the flattened surface (Figure 6b,c). These results suggest that deposited GO enabled the stable formation of a thin polyamide layer due to the hydrophilic and flatter surface, whereas the morphology of ESC-GO 1 shows sharp wrinkles between the GO agglomerates (Figure 6d). The high content of GO in ESC-GO 1 formed a cracked surface consisting of GO agglomerates with an apparently large number of defects on the polyamide layer.

### 3.3. Performance of ESC-FO Membrane

As shown in Figure 7, the ESC membrane showed reasonable osmotic water flux (15.35 and 34.53 L m^−2^ h^−1^ in 0.5 and 2 M NaCl DS) and high reverse ion flux (4.34 and 0.05 mol m^−2^ h^−1^ in 0.5 and 2 M NaCl DS) in proportion to the NaCl concentration of DS. At high ion flux, the ESC membrane is less effective than the FO membranes based on polyamide [7] and cellulose acetate/triacetate (CA/CTA) [34,35]. However, compared to the bare ESC membrane, the reverse ion flux is ~4 times lower and the pure water flux is considerably enhanced for the ESC-GO 0.1 and 0.5 membranes. The water flux was highest (19.2, 46.19 L m^−2^ h^−1^ in 0.5, 2 M NaCl DS) and the reverse ion flux (1.04, 2.01 mol m^−2^ h^−1^ in 0.5, 2 M NaCl DS) was lowest for the ESC-GO 0.5 membrane. In contrast, the lowest water flux and highest reverse ion flux were obtained with ESC-GO 1. These results indicate that the ESC-GO 1 membrane does not reject ions effectively.

Before GO coating, as shown in Figure 5, the diameter of the protein fibers in the ESM was 0.3–2 µm [36] with a wide aperture (~4 µm) between the fibers. A number of round defects were found in the polyamide layer located at the aperture between the protein fibers in the bare ESC membrane (Appendix A). These defects account for the larger reverse ion flux as they enable free ion permeation. This free ion transport reduces the osmotic gradient across the active layer, which also reduces the osmotic water flux. Similarly, the inferior performance of the ESC-GO 1 membrane could be explained in terms of the surface morphology observed by FESEM. In the ESM-GO 1 membrane, there were deep wrinkles between each GO flake, and the polyamide layer was not deposited inside the wrinkles (Appendix A). Furthermore, large GO agglomerates were found on the surface of the ESC-GO 1 membrane (Appendix A), which may have hindered stable formation of the polyamide layer. It is also known that these GO aggregates cause the formation of physical defects in the polyamide layer [37]. Therefore, the low FO performance of ESM and ESM-GO 1 is due to structural problems such as defect and partial absence in the polyamide layer. However, these defects were not found in the ESC-GO 0.1 and 0.5 membranes containing GO (Figure 6). The reduced defects could be explained by the morphology shown in Figure 4b,c. In these membranes (ESM-GO 0.1 and 0.5), coated GO flakes filled the spaces between the protein fibers. The relatively smooth and hydrophilic surface formed by GO, instead of voids between the protein fibers, may lead to stable polyamide deposition. Moreover, GO contains a large amount of hydroxyl (-OH) and carboxyl (-COOH) groups that enhance the adhesion with polyamide and are known to be compatible with polyamide [29,38,39].

Consequently, the parametric performance of FO membranes were characterized by the analytical method developed by Tiraferri et al. [26]. The coefficient of determination (*R*^2^) must be greater than 0.95 to be reliable, which is the yield value, but *R*^2^ was smaller than the yield value in the case of ESC. The low *R*^2^ value of the ESC, indicating that the determined value is not reliable, is expected due to poor and irregular ion exclusion [6], whereas the defective performance of ESC-GO 1 membrane was presented by poor parametric performance especially at dramatically high ion permeability, with reliable *R*^2^ value. The parametric performance of the ESC-GO 0.1 and 0.5 membranes were reliably calculated in valid range, with *R*^2^ higher than the yield values (Table 1). The characterized pure water permeability of the ESC-GO 0.1 and 0.5 membranes was 1.39 and 1.34 L m^−2^ h^−1^ bar^−1^, respectively. The slightly lower pure water permeability of the latter is attributed to the higher content of GO at the interface of the polyamide layer. Instead, the ion permeability was lower for ESC-GO 0.5 (3.10 L m^−2^ h^−1^) compared to ESC-GO 0.1 (3.57 L m^−2^ h^−1^). In ESG-GO 0.1, 0.5 membrane, the flatter surface structure and presence of GO may have effected a lower ion permeability by enhancing the integrity of the polyamide layer. Finally, the calculated structural parameter of the ESC FO membranes was as low as 139 µm (ESC-GO 0.1) and 125 µm (ESC-GO 0.5). The slight difference between the structural parameter of the ESC-GO 0.1 and ESC-GO 0.5 membranes is attributed to the support layer hydrophilicity promoting complete wetting [40]. Although these values are higher than that of the electro-spun polymeric support (~80 µm) [12,18], they are still considerably lower than those of membrane supports based on polymers and polymer-based nanocomposites [8,41]. These considerably low structural parameters are due to the ESM composed of crosslinked nano-microfibers constructing a less-tortuous diffusion path.

Table 2 shows the performance comparison with various FO membranes such as thin film nanocomposite (TFN), TFC, aquaporin and CA/CTA. Under similar conditions using 1 M NaCl DS in FO mode, biocomposite membranes exhibit higher water fluxes than TFC, aquaporin and CA/CTA-based membranes. The low structural parameter of biocomposite membrane, through short diffusion path between eggshell fibers in ESM, resulted in significantly higher FO performance as well as the feasibility of ESM as a material for the FO membrane.

## 4. Conclusions

In this study, we proposed a new approach for the design and fabrication of an FO membrane by exploiting the novel microstructure of ESM as a support layer. ESM was successfully employed to construct a biocomposite FO membrane by hybridization with a thin polyamide layer. Although bare ESM undesirably leads to poor integrity of the deposited polyamide layer, coating a small amount of GO on the ESM improved the integrity of the polyamide layer. Furthermore, the inherent fibrous structure of ESM, which is reported to provide a short diffusion length due to the superior porosity and interconnectivity, resulted in a low structural parameter of 138 μm [12,49]. The osmotic water flux of the biocomposite membrane reached 46.19 L m^−2^ h^−1^ with 2 M NaCl DS, because the structural parameter restricting the upper limit of the water flux was small [12,50]. The biocomposite membrane showed the potential of replacing synthetic polymer materials that occupy most of the weight of the FO membrane through the reuse of waste biomaterial. If additional scalable technology can be developed for large-area production, this approach provides green production fabricating a new type of membrane employing ESM, with significantly high FO performance as well as reduced chemical waste release.

## Figures and Tables

**Figure 1 membranes-12-00166-f001:**
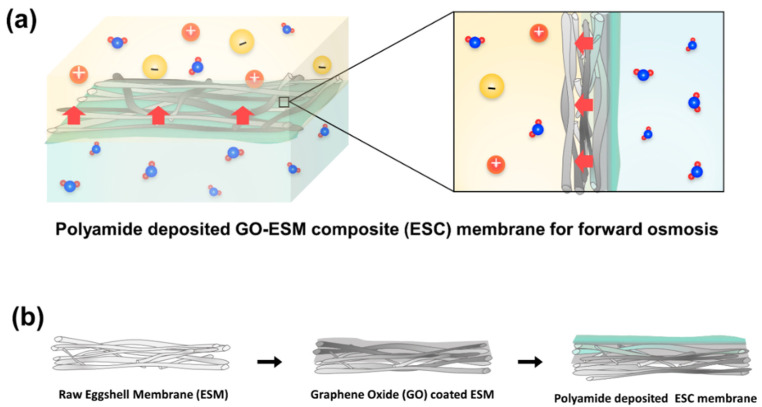
Schematic of (**a**) composite membrane for FO and (**b**) fabrication process.

**Figure 2 membranes-12-00166-f002:**
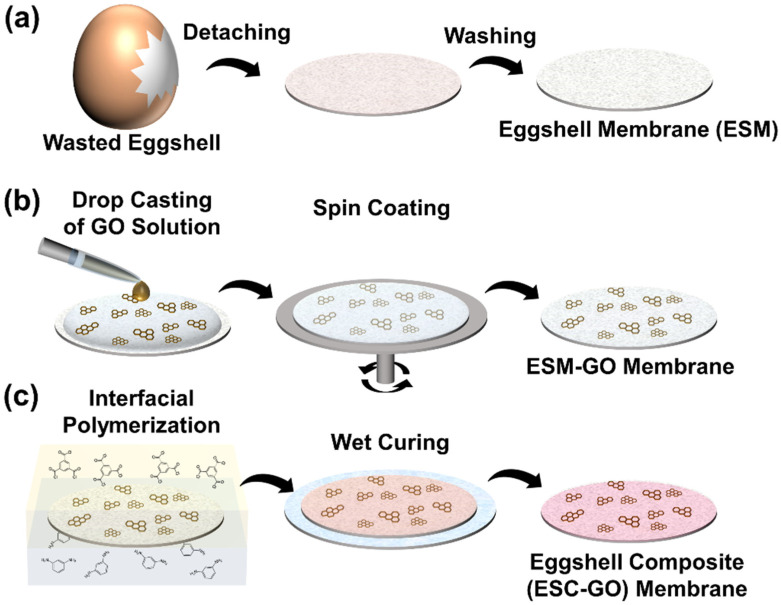
Schematic of biocomposite membrane fabrication process: (**a**) ESM preparation, (**b**) GO coating and ESM-GO membrane, (**c**) active layer fabrication and ESC-GO membrane.

**Figure 3 membranes-12-00166-f003:**
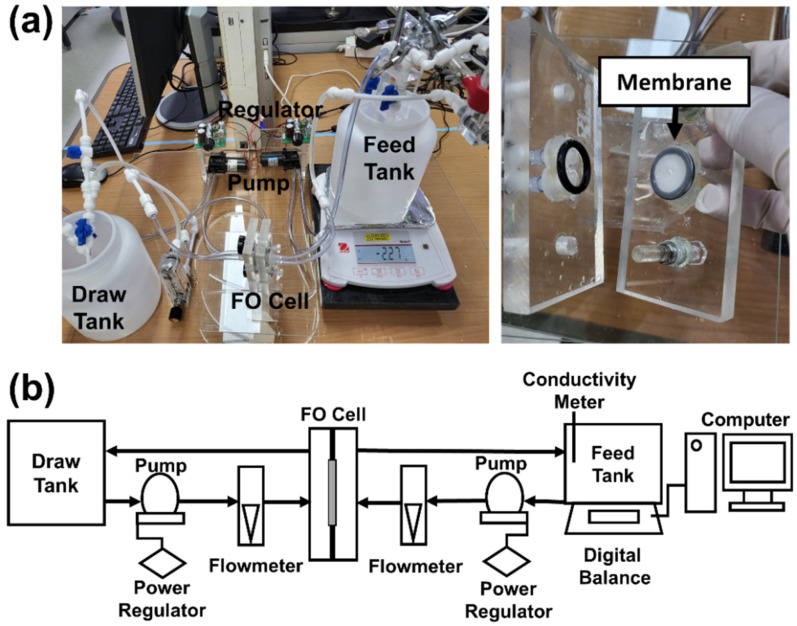
(**a**) Photograph and (**b**) schematic diagram of lab-scale FO experimental setup.

**Figure 4 membranes-12-00166-f004:**
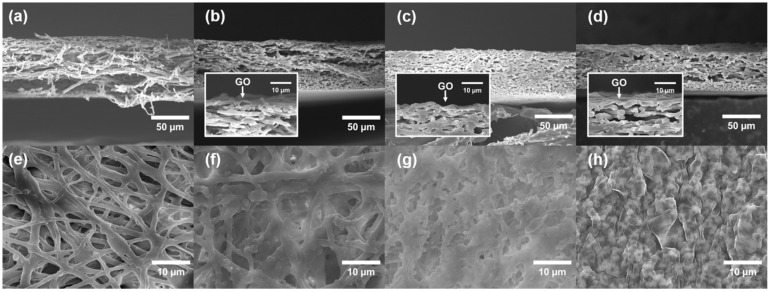
FESEM image of (**a**–**d**) top and (**e**–**h**) cross-surface of ESM (**a**,**e**) and ESM-GO composite membranes: ESM-GO 0.1 (**b**,**f**), ESM-GO 0.5 (**c**,**g**), and ESM-GO 1 (**d**,**h**).

**Figure 5 membranes-12-00166-f005:**
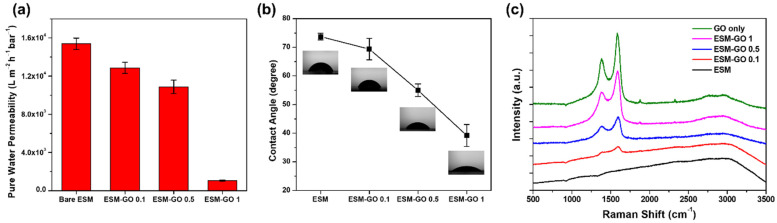
(**a**) Pure water permeability, (**b**) contact angle, and (**c**) Raman spectra of top surface of membrane.

**Figure 6 membranes-12-00166-f006:**
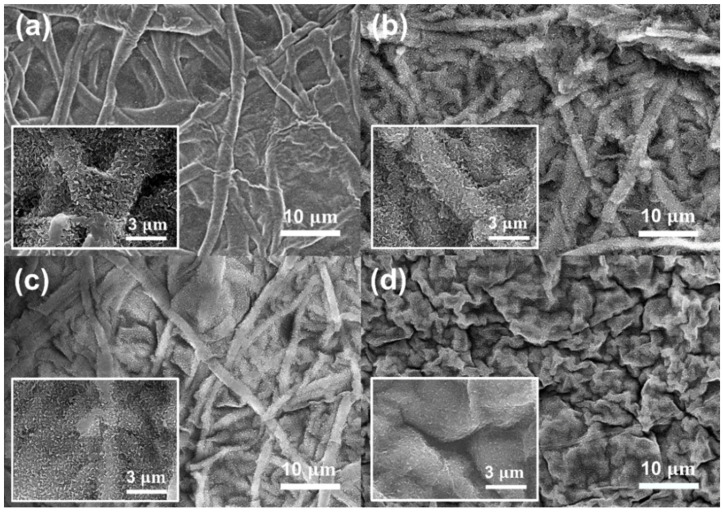
Morphology of FO membranes: (**a**) ESC, (**b**) ESC-GO 0.1, (**c**) ESC-GO 0.5, and (**d**) ESC-GO 1.

**Figure 7 membranes-12-00166-f007:**
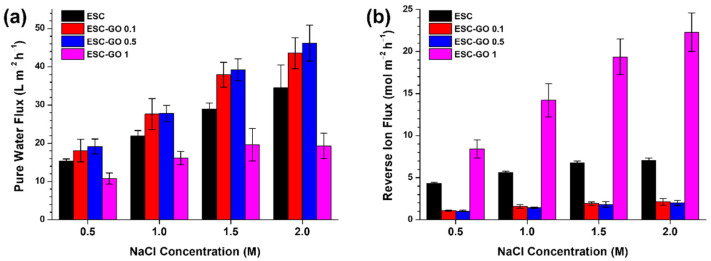
(**a**) Pure water flux and (**b**) reverse ion flux of FO membranes.

**Table 1 membranes-12-00166-t001:** Parametric performances of biocomposite membranes.

Membrane	*A* (L m^−2^ h^−1^ bar^−1^_)_	*B* (L m^−2^ h^−1^)	*S* (µm)	*R*^2^ [*J_w_*]	*R*^2^ [*J_s_*]
ESC	8.16	93.08	321	0.957	0.881
ESC-GO 0.1	1.39	3.57	138	0.981	0.972
ESC-GO 0.5	1.34	3.10	125	0.972	0.964
ESC-GO 1	7.18	355.15	243	0.997	0.997

**Table 2 membranes-12-00166-t002:** Comparison of FO performance with various types of membranes.

Type of FO Membrane	FO Water Flux (L m^−2^ h^−1^)	Structural Parameter (µm)	Draw Solution	Feed Solution	References
Biocomposite membrane	27.8	125	1 M NaCl	DI Water	This work
TFN	46	80	1 M NaCl	DI Water	[12]
TFN	24.5	351	1 M NaCl	10 mM NaCl	[42]
TFN	19.6	646	1 M NaCl	20 mM NaCl	[43]
TFC	25	312	1 M NaCl	DI Water	[6]
TFC	20	238	1 M NaCl	DI Water	[44,45]
Aquaporin	8.8	569	1 M NaCl	DI Water	[45]
Aquaporin	23.1	420	1 M NaCl	DI Water	[46]
CA/CTA	5.1	54	1 M NaCl	DI Water	[47]
CA/CTA	7.9	639	1 M NaCl	DI Water	[48]

## Data Availability

Data is contained within the article and Appendix A.

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
