# Peer review of "Thin Film Biocomposite Membrane for Forward Osmosis Supported by Eggshell Membrane"

_membranes, 2022, doi:10.3390/membranes12020166_

Round 1

Reviewer 1 Report

This is an interesting work to fabricate FO membrane by using the eggshell membrane as a bio-substrate. I recommend it to be published before a major reversion.

  1. The Introduction should further clarify the problem of the FO membranes prepared by the synthetic polymers, and then emphasize the advantages of the eggshell membranes.
  2. Several related references should be cited, such as Desalination 421 (2017) 160–168, Polymers 2019, 11, 879, Acta Polymerica Sinica, 2020, 51(4): 385-392, et al.
  3. In the Experimental section, it is lack of the preparation detail about the GO coating. What’s the method and the specific parameters? (Page 3, lines 81-82)
  4. How to calculate the coefficient of determination (R2)? It should be clarified in the Experimental section.
  5. Some errors should be revised: The repeated paragraph in page 3; Page 4, lines 145-147: “After deposition of GO, the top surfaces of the membrane were gradually covered by GO (Figure 2f,g) with increasing GO content, a thick GO layer fully covered the top surface of the ESM-GO 1 (Figure 2d,h).”
  6. It is satisfactory that the ESM-GO0.1 and 0.5 display the increasing water flux and the decreasing reverse salt flux. Generally, the increasing water flux is companied with an increasing reverse salt flux. The reason is not sufficient in this paper, and further characterization should be added. For example, the thickness of PA layer can be detected by using SEM or TEM. The crosslinking degree of PA layer can be determined by XPS.

Reviewer 2 Report

An interesting study by Kim et al. showed the preparation of a thin film composite membrane for forward osmosis using an eggshell membrane. The effect of membrane support functionalization with GO was also examined. The performance of the prepared membranes was good. Overall, the structure of the manuscript is logical. Having said that, the authors need to revise the manuscript extensively to make it fit for publication.

  1. Section 2.2 to 2.3: It would be good to provide a schematic diagram depicting the various steps in the membrane preparation.
  2. A photograph of the FO test setup with the details of the experimental conditions should be provided in section 2.4.
  3. The authors should provide more details, e.g., operating pressure, filtration duration, compaction (if any), and so on, about the filtration experiment conditions. How did they perform the experiments so that others could reproduce them?
  4. Did the authors control the thickness of the GO layer deposited on the ESM membrane?
  5. Did they observe any change in the coating layer thickness with the varying GO concentration used for the coating purposes?
  6. Did the authors observe any effect of GO aggregation on the membrane properties?
  7. Surface roughness plays a crucial role in the separation performance of the membranes. Therefore, membrane roughness data are required. It will also help understand the arguments made on page 6, lines 197-206.
  8. Pure water flux increased for membranes prepared with 0.1 and 0.5 wt.% GO, whereas it suddenly dropped for membrane prepared with 1 wt.% GO. The authors should provide a convincing argument for this trend. Provide characterization data to support your arguments.
  9. Long filtration performance should be provided to convince that the membranes are stable for FO application.
  10. Further, the stability of the membranes is also missing. Provide characterization data to prove that the membranes are stable after the filtration experiments. Leaching of modifiers, e.g., GO, should also be discussed.
  11. A table comparing the performance of this study to that reported in the literature should also be provided. All critical parameters, e.g., membrane composition, filtration experiment conditions, flux, etc., should be considered for comparison purposes.
  12. Conclusions should include more quantitative information.

Round 2

Reviewer 1 Report

None

Reviewer 2 Report

The revised version is better. It can be accepted for publication.